# Seroepidemiology of human leptospirosis in the Dominican Republic: A multistage cluster survey, 2021

Eric J. Nilles [1,2,3]*, Cecilia Then Paulino[4], Renee Galloway[5], Michael de St. Aubin[1,3], Helen J. Mayfield[6], Angela Cadavid Restrepo[6], Devan Dumas[1,3], Salome Garnier[1,3], Marie Caroline Etienne[1], William Duke[7], Farah Peña[4], Naomi Iihoshi[1], Gabriela Abdalla[1], Beatriz Lopez[8], Lucia de la Cruz[4], Bernarda Henríquez[4], Kara Durski[1,3], Margaret Baldwin[1,3], Gideon Loevinsohn[1,9], Eleanor M. Rees[10], Beatris Martin[6], Benn Sartorius[6], Ronald Skewes-Ramm[4], Emily Zielinski Gutiérrez[8], Adam Kucharski[11], Colleen L. Lau[6]

**1** Brigham and Women's Hospital, Boston, Massachusetts, United States of America, **2** Harvard Medical School, Boston, Massachusetts, United States of America, **3** Harvard Humanitarian Initiative, Cambridge, Massachusetts, United States of America, **4** Ministry of Health and Social Assistance, Santo Domingo, Dominican Republic, **5** Division of High-Consequence Pathogens and Pathology, Centers for Disease Control and Prevention, Atlanta, Georgia, United States of America, **6** University of Queensland Centre of Clinical Research (UQCCR), University of Queensland, Brisbane, Australia, **7** Pedro Henríquez Ureña National University, Santo Domingo, Dominican Republic, **8** Centers for Disease Control and Prevention, Central America Regional Office, Guatemala City, Guatemala, **9** Massachusetts General Hospital, Boston, Massachusetts, United States of America, **10** University of Glasgow, Glasgow, United Kingdom, **11** London School of Hygiene & Tropical Medicine, London, United Kingdom

☯ These authors contributed equally to this work.
* enilles@bwh.harvard.edu

## Abstract

Little is known about the epidemiology of leptospirosis in the Dominican Republic, the second most populous country in the Caribbean. We report on findings from a multi-stage household survey across two regions in the country that reveals a previously under-estimated burden of human *Leptospira* infection. Our findings, based on the reference-standard microscopic agglutination test, indicate a complex picture of serogroup diversity, spatial heterogeneity in infection and risk, and a marked discrepancy between reported cases and serologically estimated infections. Given an overall seroprevalence of 11.3% (95% CI: 10.8–13.0%) and a lower estimated force of infection (0.30% per year [0.27%–0.35%]) the number of infections may exceed national reported case data by 145-fold or more. Icterohaemorrhagiae, associated with severe Weil's disease, was the most commonly identified serogroup with a serogroup-specific prevalence of 4.4%. Consistent with other settings, risk factors including age, male sex, and rat exposure were associated with higher seroprevalence. Our study highlights the need for targeted public health inter-ventions informed by serogroup-specific dynamics, detailed spatial analyses, knowledge of local animal reservoirs, and strengthened laboratory surveillance to effectively control this pathogen.

**Data availability statement:** A complete de-identified dataset and R code are available at https://github.com/enilles1/DR-Leptospirosis for the purpose of reproducing and building on the analyses.

**Funding:** This work was primarily funded by the Centers for Disease Control and Prevention (U01GH002238 to EJN, CLL and AK). Salary support was provided by the Wellcome Trust (206250/Z/17/Z to AK) and the Australian National Health and Medical Research Council (APP1158469 to CL). The funders, specifically the CDC, were involved in the design of the study, data collection, analysis, and interpretation of data. Additionally, CDC staff contributed to the manuscript's editing process. The Wellcome Trust and the Australian National Health and Medical Research Council did not have direct involvement in the study's design, data collection, analysis, interpretation of data, or manuscript preparation.

**Competing interests:** I have read the journal's policy and the authors of this manuscript have the following competing interests: EJN is the PI on a US CDC funded U01 award that funded the study, and CLL, AK, DD, MdSA, ACR, HM, SG, MCE, WD, NI, GA, BH, KD, and MB have received salaries, consultancy fees, or travel paid through this award. EZG and BL are employees of the US CDC. BH, CT, LC, FP, and RSR are employees of the Ministry of Ministry of Health and Social Assistance, Dominican Republic, that was subcontracted with funds from the US CDC award. AK is supported by the Wellcome Trust, UK. CLL is supported by the Australian National Health and Medical Research Council. We declare no other competing interests.

## Author summary

Leptospirosis is a significant public health concern in many tropical regions, but its true burden is often unrecognized due to limited diagnostic capacity and under-reporting. Using a multistage household survey, we assessed human leptospirosis infection across two regions in the Dominican Republic. Our findings revealed a high diversity of Leptospira serogroups, regional differences in transmission, and a marked gap between reported cases and actual infections. Estimated annual infection rates were over 100 times higher than nationally reported cases, underscoring substantial under-ascertainment. We identified key risk factors for infection, including older age, male gender, and rat exposure in the home, with seroprevalence varying across regions. Notably, different Leptospira serogroups demonstrated unique transmission patterns likely linked to distinct animal reservoirs, such as rats, livestock, and dogs. For example, serogroup Australis was more common among farmers, reflecting livestock exposure. This study highlights the value of serological surveys in uncovering hidden burdens of disease and identifying tailored public health interventions, such as rodent control and animal vaccination. By understanding the complex epidemiology of Leptospira infections, our findings contribute to improved surveillance and targeted strategies to control leptospirosis in endemic settings.

## Introduction

Leptospirosis is a growing global health threat. The zoonotic disease, caused by *Leptospira* spirochetes, is transmitted to humans through direct exposure to infected animals or more commonly through contact with soil or water contaminated with urine from infected mammals. Globally, there are over one million yearly cases of leptospirosis and approximately 60,000 deaths [1]. While Asian and Oceanic nations have reported the highest incidence of the disease to date, leptospirosis has become the most widespread and prevalent bacterial zoonotic disease globally, and its epidemiological reach spans tropical and subtropical areas, including many regions with limited laboratory diagnostics or well-established surveillance infrastructure [1,2].

The diagnosis of leptospirosis is challenging given its varied clinical presentations, ranging from asymptomatic or mild febrile infections to severe pulmonary hemorrhage and multi-organ failure, which overlap with the symptomatology of many tropical and subtropical diseases including dengue, chikungunya, typhoid, rickettsioses, and malaria. The accurate diagnosis relies on high clinical suspicion coupled with laboratory evidence of infection. While molecular tests have excellent test characteristics for the diagnosis of infection early in the disease course, they are costly and not widely available in many endemic settings including in the Dominican Republic, the location of the present study. The reference standard serologic assay is the microscopic agglutination test (MAT), that detects antibodies against leptospiral lipopolysaccharide. Anti-*leptospira* antibodies are generated within one week of infection and persist for up to eight years. While the utility of MAT is limited for the diagnosis of acute leptospirosis due to poor sensitivity early in the disease course, MAT provides valuable data on the presence of *leptospira* antibodies for population-based serological surveys [3,4].

Given the overlapping clinical features of leptospirosis and other common diseases, together with limited access to confirmatory testing, the burden and epidemiology of leptospirosis is poorly characterized in the Dominican Republic and the Caribbean more broadly [5]. Yet, despite this, the Caribbean region is known to have widespread endemic disease, has

experienced numerous outbreaks, and reports among the highest estimated disease-specific mortality rates globally [1]. Regional risk factors for leptospirosis transmission include intensive farming and household livestock, as well as frequent extreme climactic events with excess precipitation and flooding, which are exacerbated by climate change and expected to increase [6,7]. In the present study, to inform public health planning and control efforts, we aimed to characterize the seroepidemiology of human *leptospira* infection among populations in two provinces in the Dominican Republic.

## Methods

### Ethics statement

Written consent was obtained for all participants. For children <18 years old, except emancipated minors, written consent was obtained from the legal guardian. Written assent was provided by adolescents 14–17 years old, and verbal assent by children 7–13 years old. The study protocol was approved by the National Council of Bioethics in Health, Santo Domingo (013-2019), the Institutional Review Board of Pedro Henríquez Ureña National University, Santo Domingo, and the Mass General Brigham Human Research Committee, Boston, USA (2019P000094). Study procedures and reporting adhered to STROBE criteria for observational studies.

### Setting

The Dominican Republic, a Caribbean nation with a population of almost 11 million residents, is an upper middle income Latin American country that shares the island of Hispaniola with Haiti. Case reports and outbreak investigations have documented case reports among returning travelers and intermittent surges in leptospirosis cases following tropical storms and hurricanes [7,8]. The present study is set in the provinces of Espaillat and San Pedro de Macoris, selected to ensure spatial, ecological, and geographical heterogeneity. Espaillat Province, located in the northwest of the country (Northwest study region), has a population of about 260,000 residents with topography ranging from coastal plains to mountainous areas. The climate is tropical with annual rainfall of 1,300 mm, and a rainy season from June to November. Farming activities include coffee, cacao, and rice cultivation. San Pedro de Macorís Province has a population of about 420,000 and is a coastal province located in the southeast of the country, with topography characterized by a coastal plain that houses most of its population and economic activities (Southwest study region). The province is historically recognized as a center for sugar cane farming. It receives about 1,600 mm of rain annually, and is prone to tropical storms, particularly during the June to November rainy season.

### Study design and participant selection

As previously described [9], between June 30 and Oct 10, 2021, we conducted a three-stage cross-sectional national household serological survey and enrolled 6,683 participants from 3,832 households in 134 clusters across all 32 provinces nationally. Briefly, to maximize spatial distribution we used grid- and satellite-image based methods to select clusters and 60 households within each cluster [10]. Household members aged ≥5 years old present in the home at the time of the serosurvey were invited to participate. The present study is a nested analysis of 23 clusters in Espaillat (n = 10) and San Pedro de Macoris provinces (n = 13). These provinces were intentionally oversampled during the national survey to allow for more granular provincial analyses.

## Study procedures and immunoassay characteristics

Questionnaires were administered to all study participants by trained enumerators using the KoBo Toolbox data collection platform (www.kobotoolbox.org) on electronic tablets to collect self-reported demographics (age, gender, race-ethnicity), education, primary occupation, work location (indoors, outdoors, or a mix), rural vs. urban residence, residence in a barrio (settlement), and seeing rats in their residence. Venous blood was collected from all participants, processed as sera, and frozen at −80 °C. MATs were used to detect anti-*Leptospira* antibodies, and determine the putative serogroups associated with infections. Serological analyses were performed at the US CDC's Zoonoses and Select Agent Laboratory, Bacterial Special Pathogens Branch, Atlanta, USA. A panel of 20 pathogenic serovars comprising 17 serogroups were selected for the MAT panel (S1 Table), and samples were tested at dilutions from 1:100 to endpoint. MAT titers ≥1:100 were considered seropositive and indicative of prior infection.

## Data sources

National and provincial demographic data including cluster population and classification (urban versus rural) were provided by the Dominican Republic National Statistics Office and the United Nations Statistics Division [11,12]. Data on national reported leptospirosis cases was from the Pan American Health Organization (PAHO) Core Indicator Dashboard from data generated by the General Epidemiology Directorate of the Ministry of Public Health and Social Assistance. Leptospirosis is a reportable disease in the Dominican Republic. Other data were enumerated during the study.

## Classification and statistical analysis

Given cross-reactivity between serovars, particularly those in the same serogroup, analyses were performed on the serogroup level. For samples that reacted to multiple serogroups, the serogroup associated with the highest titer was considered to be the primary reacting serogroup. Samples that registered the same highest titer across two or more serogroups were recorded as 'mixed' (i.e., reacting to multiple serogroups). Serogroups with less than ten sero-reactive samples were aggregated and reported as 'other.' Adjusted seroprevalence estimates were weighted for sampling design (selection probability, clustering), corrected for finite population, and post-stratified for age group and sex using methods previously described [13].

To assess risk factors for infection, logistic regression analysis was performed with serostatus as the dependent binary variable, with only those variables with p-value <0.05 on univariable analysis retained for the multivariable models. Explanatory variables explored in the models were age category, gender, study region, setting, occupation, and frequent contact with rats. To explore serogroup-specific risks, we used the same models but adjusted the dependent variable to represent only those individuals with the highest titer to the relevant serogroup as the serogroup-specific seropositives, with all others representing the serogroup-specific seronegatives.

Serocatalytic models were used to estimate the annual force of infection (FOI) from serological survey. A catalytic model (model 1), which assumes there is a lifelong constant FOI, and a reverse catalytic model (model 2a), which assumes that antibody prevalence declines over time, were fitted to the serological survey. In addition, a third model (model 2b) which is a reverse catalytic model which assumes no transmission for the six years prior to the serosurvey was also explored. A uniform distribution between 0 and 0.5 (corresponding to a yearly attack rate between 0 and 39%) was used for the FOI. Informed priors were used for the rate

of waning with a distribution mean of 0.14 which corresponds to a duration of antibody body positivity of ~7 years and is informed based on previous estimates (S2 Table) [3].

Analyses and data visualization were performed using the R statistical programming language (R version 4.2.3, 2023-03-01) [14], with the *RANN* package for cluster and household selection [15], the *survey* package for the analysis of complex survey samples [16], *finalfit* (glm) for logistic regression [17], *mgcv* for partial effect analysis, and *ggplot2* for data visualization [18].

## Results

Of 2,124 participants enrolled in the two study provinces during the national serosurvey, 2,091 samples (98.4%) were tested for evidence of prior leptospirosis infection using MAT and comprise the current study population. The median age was 39 years (IQR 22, 56) and 1,339 were female (64.0%). Demographic details are listed in Table 1. A total of 237 study participants registered agglutinating antibodies, assessed by the presence of MAT titers ≥1:100, for an overall unadjusted seroprevalence of 11.3% (95% confidence interval: 10.0–12.7%) and an adjusted seroprevalence of 11.3% (10.8–13.0%).

Serocatalytic models estimated the annual attack rate in the Dominican Republic to be 0.30% (0.27–0.35%) using the catalytic model (model 1) which assumes a constant FOI and no waning immunity, and 1.00% (0.70%–1.69%) for the reverse catalytic model (model 2a) which assumes a constant FOI whilst allowing for waning immunity. The catalytic model (model 1) fitted the data better compared with the reverse catalytic model (model 2a), with a DIC difference of 31. A third model, which was the reverse catalytic model but assumed there was no transmission in the six years prior to the serosurvey (model 2b), fitted the data better compared with model 2a (DIC difference 16), with an estimated annual attack rate of 1.19% (0.90–0.17%) (S2 Table and S1 Fig). Using data from these models, we estimated the number of annual leptospirosis infections and assessed the gap between reported cases and serologically inferred infections. Using the lowest estimated FOI (0.30%) applied to the population of the Dominican Republic (~11 million), we infer about 33,000 infections per year which is approximately 145-fold higher than the average number of nationally reported cases between 2013 and 2022 (226 cases per year) (S3 Table). If we consider the highest estimated FOI (1.19%), the ratio of reported to serologically estimated cases is approximately 1:580. While the spatial heterogeneity of leptospirosis transmission necessitates caution when extrapolating from two provinces to the entire country, when considering only the two study provinces (population ~680,000) we estimate 2,040 annual leptospirosis cases assuming a FOI of 0.30% and 8,092 cases assuming a FOI of 1.19%. In turn, this suggests that there may be between nine and 36-fold more leptospirosis infections in just the two study provinces than reported cases across the entire country.

Seropositive results were reported for 16 of the 20 serovars included in the MAT panel, which included 14 of 17 serogroups. Among the 237 seropositive individuals, 91 (38.4%) registered the highest titers to *L. interrogans* serogroup Icterohaemorrhagiae, 38 to Australis (16.0%), 25 to Canicola (10.5%), 20 to Djasiman (8.4%), and 12 to *L. santarosai* serogroup Pyrogenes (5.1%) (Fig 1 and Table 2 and S1 Table). The four most common serogroups accounted for 73.4% of seropositive results. Among 27 (11.4%) individuals that registered highest titers against two or more serogroups (i.e., Mixed serogroups), the mostly commonly reacting serogroups were Icterohaemorrhagiae (23, 85.2%), Canicola (14, 51.9%), and Australis (12, 28.6%). Most seropositive individuals registered titers of 1:100 (N = 119, 50.2%), 1:200 (49, 20.7%), 1:400 (44, 18.6%), and 1:800 (14, 5.9%), with 11 (4.6%) registering titers of 1:3200 or higher (Fig 1). The distributions of titers when stratified by serogroup were largely similar (Fig 1).

**Table 1. Characteristics of study population and *Leptospira* seroprevalence by microscopic agglutination test, Espaillat and San Pedro de Macoris Provinces, Dominican Republic, 2021.**

| Population characteristic | N | No. seropositive | Seroprevalence - unadjusted percent (95% CI) | Seroprevalence - adjusted percent (95% CI) |
|---|---|---|---|---|
| Overall | 2091 | 237 | 11.3 (10.0, 12.7) | 11.3 (10.8, 13.0) |
| **Gender** | | | | |
| Female | 1339 | 103 | 7.7 (6.3, 9.2) | 8 (6.8–9.3) |
| Male | 736 | 131 | 17.8 (15, 20.7) | 15.9 (12.9–19.2) |
| Other | 16 | 3 | 18.8 (0.5, 42.6) | 9.8 (5.6–15.6) |
| **Age category, years** | | | | |
| 5–19 | 395 | 11 | 2.8 (1.2, 4.4) | 4.2 (2.4, 7.1) |
| 20–34 | 531 | 51 | 9.6 (7.1, 12.1) | 12.6 (9.8, 16.0) |
| 35–49 | 461 | 63 | 13.7 (10.5, 16.8) | 10.7 (4.6,23.0) |
| 50–64 | 393 | 52 | 13.2 (9.9, 16.6) | 16.1 (14.1, 18.4) |
| 65+ | 311 | 60 | 19.3 (14.9, 23.7) | 19.4 (15.6, 23.9) |
| **Study region** | | | | |
| Northwest (Espaillat) | 811 | 127 | 15.7 (13.3, 18.3) | 14.9 (13.9–15.9) |
| Southeast (San Pedro de Macorís) | 1280 | 110 | 8.6 (7.1, 10.1) | 7.1 (4.1–11.1) |
| **Setting** | | | | |
| Rural | 911 | 128 | 14.1 (11.8, 16.3) | 13.6 (7.3–22.1) |
| Urban | 1180 | 109 | 9.2 (7.6, 10.9) | 11.5 (8.1–15.6) |
| **Reside in barrio** | | | | |
| No | 1527 | 173 | 11.3 (9.8, 13.0) | 11.6 (9.7–13.8) |
| Yes | 541 | 62 | 11.5 (9.0, 14.4) | 13.7 (11.3–16.6) |
| **Work setting** | | | | |
| Indoor | 175 | 25 | 14.3 (9.6, 19.9) | 10.6 (3.7–22.1) |
| Outdoor | 92 | 20 | 21.7 (14.1, 30.9) | 21.5 (6.4–45.4) |
| Mix | 290 | 50 | 17.2 (13.1, 21.8) | 14.2 (9.4–20.2) |
| **Education** | | | | |
| No formal | 208 | 43 | 20.7 (15.7, 26.7) | 21.0 (17.6–24.7) |
| Primary | 627 | 104 | 16.6 (13.9, 19.7) | 21.2 (17.5–25.3) |
| Secondary | 772 | 66 | 8.5 (6.7, 10.7) | 7.9 (5.5–11.1) |
| Technical | 44 | 5 | 11.4 (4.8, 24.6) | 18.3 (11.9–27.0) |
| University | 204 | 15 | 7.4 (4.4, 24.6) | 5.7 (1.5–19.5) |
| **Occupation** | | | | |
| Farmer | 74 | 26 | 35.1 (24.8, 46.5) | 37.0 (22.9–52.8) |
| Housewife/husband | 562 | 54 | 9.6 (7.3, 12.2) | 11.8 (9.6–14.3) |
| Non-professional | 185 | 29 | 15.9 (11.0, 21.6) | 9.9 (3.5–20.7) |
| Professional | 73 | 9 | 12.3 (6.0, 21.2) | 9.3 (0.6–37.7) |
| Retired | 64 | 14 | 21.9 (12.9, 33.1) | 26.6 (15.2–40.6) |
| Student | 420 | 13 | 3.1 (1.7, 5.0) | 4.3 (2.3–7.2) |
| Other | 713 | 92 | 12.9 (10.6, 15.5) | 13.1 (11.6–14.6) |
| **Contact with rats** | | | | |
| No | 1757 | 196 | 11.2 (9.7, 12.7) | 12.2 (10.7–13.7) |
| Yes | 332 | 41 | 12.3 (9.0, 16.2) | 10.6 (7.4–14.4) |

Missing data includes 25 for Reside in barrio (adjusted seroprevalence 7.6% (6.3–8.8)), and two for Contact with rats. Work environment enumerated for active workers and excluded students, housepersons, retirees and unemployed (adjusted prevalence for excluded: 10.8% (9.1–12.6)). No values were missing for other covariates. Adjusted seroprevalence weighted for study design (selection probability, clustering), finite population correction, and post-stratified for age and sex. Seropositive defined as reactive on MAT ≥1:100.

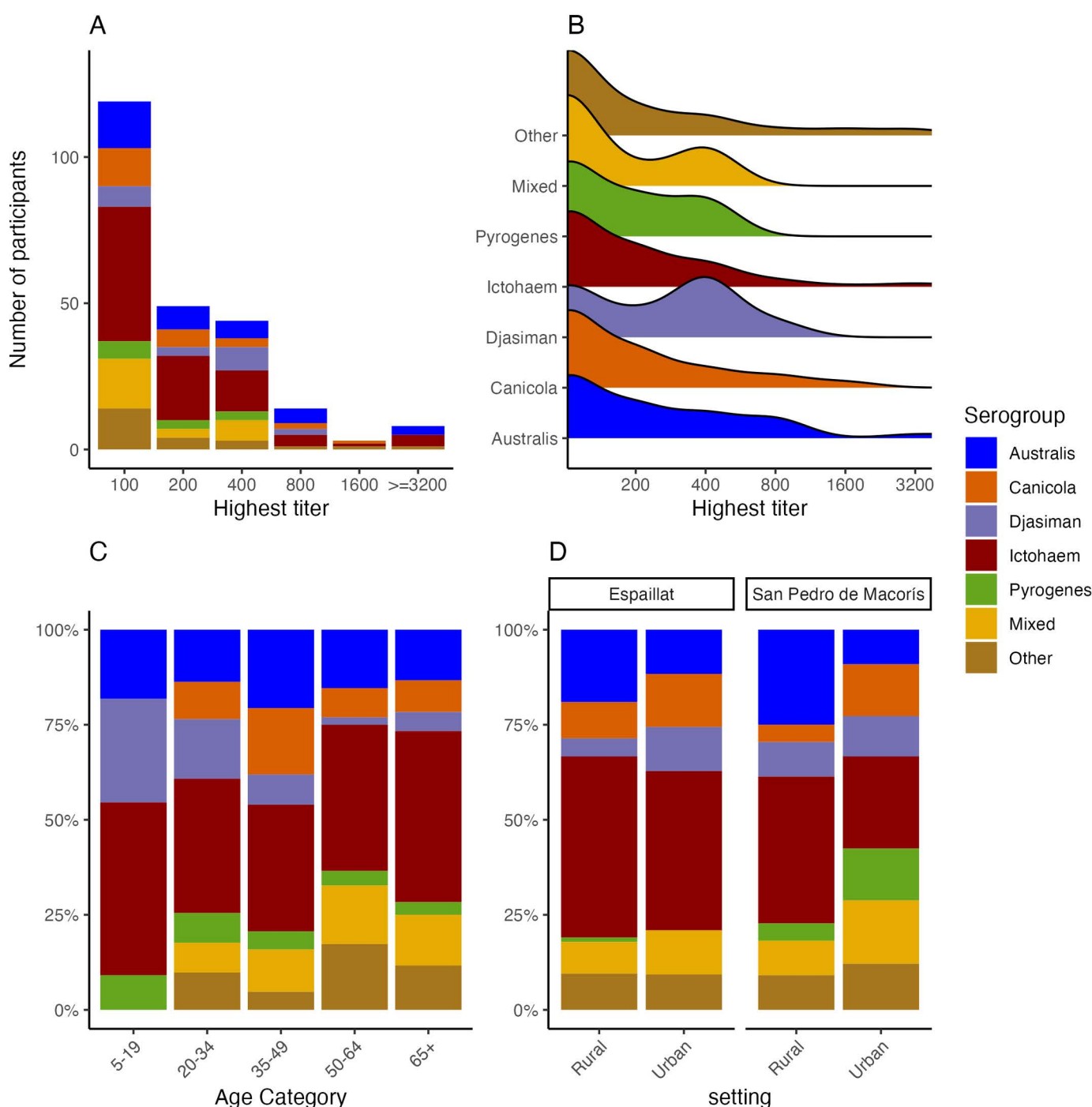

**Fig 1. Distribution of *Leptospira* serogroups by highest titer, age and study setting, Dominican Republic, 2021.** (A) Bar plot indicates the number of participants stratified by the highest MAT titer and serogroup. (B) Ridge plot shows the distribution of MAT titers for each serogroup. (C) Stacked bar plot indicates the distribution of serogroups by age group. (D) Distribution of serogroups stratified by urban vs rural setting and Northwest (Espaillat) vs Southeast (San Pedro de Macoris) regions. Serogroup is defined as the serogroup that registered the highest MAT titer for each seropositive study participant. Individuals that registered the same highest titer to two or more serogroups are categorized as 'Mixed.' N = 237 for all plots. Serogroups aggregated as 'Other' are those with counts less than 10, with details listed in S1 Table.

**Table 2.  Prevalence of positive *Leptospira* microscopic agglutination tests by primary reacting serogroup, Espaillat and San Pedro de Macoris Provinces, Dominican Republic, July-Oct 2021.**

| Serogroup | No. seropositive | Prevalence, % | Proportion of seropositives, % |
|---|---|---|---|
| Icterohaemorrhagiae | 91 | 4.35 | 38.4 |
| Australis | 38 | 1.82 | 16.0 |
| Mixed | 27 | 1.29 | 11.4 |
| Canicola | 25 | 1.20 | 10.5 |
| Djasiman | 20 | 0.96 | 8.4 |
| Pyrogenes | 12 | 0.57 | 5.1 |
| Pomona | 5 | 0.24 | 2.1 |
| Ballum | 4 | 0.19 | 1.7 |
| Tarassovi | 4 | 0.19 | 1.7 |
| Bataviae | 3 | 0.14 | 1.3 |
| Seroje | 3 | 0.14 | 1.3 |
| Mini | 2 | 0.10 | 0.8 |
| Autumnalis | 1 | 0.05 | 0.4 |
| Celledoni | 1 | 0.05 | 0.4 |
| Cynopteri | 1 | 0.05 | 0.4 |
| Hebdomadis | 0 | 0.00 | 0.0 |
| Grippotyphosa | 0 | 0.00 | 0.0 |
| Javanica | 0 | 0.00 | 0.0 |
|  | 237 | 11.3 | 100 |

Primary reacting serogroup is defined as the serogroup that registered the highest MAT titer for each seropositive study participant (N = 2,091). Individuals that registered the same highest titer to two or more serogroups are categorized as 'Mixed.' Seropositive defined as ≥1:100 titers using the MAT. Prevalence of serovars, species, and strains presented in S1 Table.

Analysis of leptospirosis seroprevalence across demographic and environmental factors identified distinct trends (Fig 2 and S4 Table). Increasing age was associated with higher seropositivity, with the seroprevalence of 2.8% among those 5–19 years, 13.7% among those 35–49 years, and 19.3% among those 65 years and older. Similarly, multivariable adjusted odds ratios demonstrated a sharp increase in seroprevalence during early adulthood, followed by a plateau in middle age and then a subsequent increase in older age (Fig 2). Seropositivity was higher among males (17.8%) than females (7.7%) (adjusted Odds Ratio 2.41 [95% confidence interval 1.79–3.25]), among residents of the Northwest (15.7%) versus Southeast Region (8.6%) (aOR 1.84, 1.31–2.58), and among those reporting frequently seeing rats in the household (aOR 1.64 [1.06–2.52]). While being a farmer was associated with high seroprevalence (35.1%) with an unadjusted odd ratio of 4.67 (2.80–7.63) vs non-professionals, no association was identified on multivariable analysis (aOR 1.63 [0.94–2.80]). Similarly, residing in a rural vs urban setting was associated with higher odds of seropositivity on univariable (OR 1.61 [1.22–2.11]) but not multivariable analysis. Residing in a barrio (settlement) and indoor versus outdoor work were not associated with leptospirosis serostatus.

Given the distinct host and ecological features of different *Leptospira* serogroups, we examined serogroup specific demographic and environmental risks. Analyzing serogroup-specific seroprevalence for the most common serogroups (Icterohaemorrhagiae, Australis, and Canicola) identified similar overall seroprevalence trends across demographic and environmental factors (Fig 2 and S4–S8 Tables). However, there were exceptions. Males registered higher seropositivity (vs females) against serogroup Icterohaemorrhagiae (aOR 2.69 [1.71–4.26]) but not Australis (aOR 1.14 [0.51–2.41]) or Canicola (aOR 1.44 [0.59–3.34]). Serogroup Australis

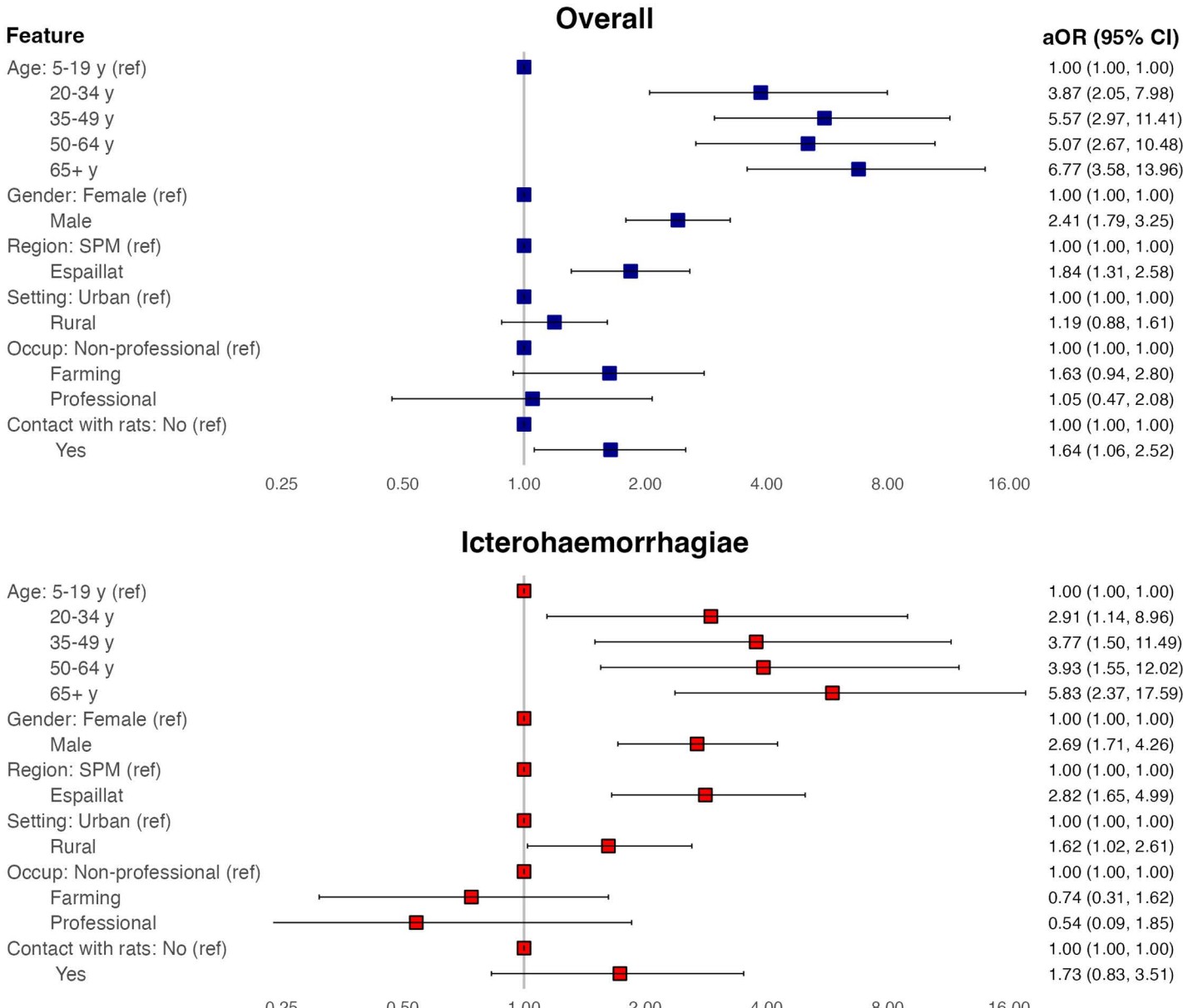

**Fig 2. Risk factors for** *Leptospira* **seropositivity by microscopic agglutination test in the Dominican Republic, 2021.** Multivariable logistic regression with all modeled covariates shown. Serogroups Australis and Canicola not presented given limited number of seropositive individuals (N = 38 and 25 respectively). Full model outputs including univariable odds ratios and model metrics shown in S4 and S5 Tables. SPM, San Pedro de Macoris (Southeast study region). Espaillat (Northwest study region).

was associated with higher seroprevalence among farmers (aOR 5.55 [2.03–14.98)] versus the reference group, but not Icterohaemorrhagiae or Canicola. Icterohaemorrhagiae and Canicola serogroups were both associated with higher seroprevalence in the Northwest study region (aOR 2.82 [1.65–4.99] and 3.24 [1.23–9.37] respectively), but not Australis. Lastly, Icterohaemorrhagiae and Canicola seroprevalence was higher in rural vs urban setting (aOR 1.62 [1.02–2.61] and 2.1 [1.0–4.7] respectively) but trended lower for Canicola (aOR 0.57 [0.23–1.35]). When excluding individuals that were seropositive to serogroups other than the serogroup of

interest (e.g., when assessing risk factors for serogroup Icterohaemorrhagiae, all seropositive individuals with highest titer to non-Icterohaemorrhagiae serogroups were excluded from the analysis) risk factors and odds ratios were essentially unchanged.

Given the unusual age distribution of seroprevalence, with the highest rates among the older population rather than the common pattern of peaking in middle age and then either plateauing or declining in older age [19,20], we examined the age distribution in more detail, stratifying by study region, gender, and urban vs. rural setting. We found that the trend of increasing seroprevalence with age was consistent across all variables, except for female gender, which showed a biphasic curve with peaks in middle age and again in older age (Fig 3). Similar trends by age group were observed when examining gender-specific seroprevalence in the Northwest vs. the Southeast region (S2 Fig). To further assess this finding, we used GAMs to model seroprevalence by age and gender, while accounting for documented or potential confounding covariates such as study region, work environment, residence in a barrio (settlement), and urban vs. rural residence. The model aligned with the unadjusted analyses showing females with a biphasic age-associated seroprevalence distribution and males with a near-linear increase in seroprevalence with age (S3 Fig).

## Discussion

This multistage household survey based on a carefully sampled population across two discrete study regions provides new insights into the epidemiology of human Leptospira infection in the Dominican Republic. Our findings suggest a wide diversity of circulating serogroups, regional variation in transmission, and a marked difference between national reported cases and infections.

Discerning the true extent of leptospirosis infections is challenging, particularly in most endemic setting where access to testing is limited. Accordingly, population-based serological studies are valuable for characterizing the burden of *Leptospira* infections. In this study we used serological data to estimate the annual attack rate and in turn the difference between nationally reported leptospirosis cases and serologically estimated infections. Our findings suggest marked under ascertainment with 100-fold or greater difference in reported clinical cases versus estimated infections. While our estimates vary based on the modeling approach it appears clear that only a small fraction of infections are identified through existing reporting mechanisms. The application of serocatalytic methods to estimate leptospirosis FOI and, in turn, under ascertainment has only been reported once [3]. In that study, based on data from a national serological survey in Fiji, the ratio of reported cases to estimated infections ranged from about 1:28 to 1:140. While surveillance infrastructure and criteria for reporting leptospirosis cases varies markedly between countries, thus making direct comparisons challenging, our findings support the concept that under-ascertainment of *Leptospira* infections in many endemic settings is likely on the order of magnitudes. The reason for such marked discrepancy between reported cases and estimated infections is likely due to a range of factors including subclinical infections or mild infections that do not seek medical care, limited access to healthcare or leptospirosis testing, or gaps in clinician knowledge about the clinical features of leptospirosis precluding suspicion of leptospirosis as the infecting pathogen.

While Icterohaemorrhagiae, Australis, Canicola, and Djasiman were collectively implicated as the infecting serogroups in almost three quarters of cases, our study suggests high diversity with 14 of the 17 serogroups included in our panel identified as the primary infecting serogroup, similar to Trinidad where 17 serovars have been isolated from humans [21]. This is in contrast to some other island settings such as Barbados where only three serovars of *Leptospira interrogans* have been identified, or Fiji where a single serovar (Pohnpei) was implicated in over 80% of seropositive cases identified through a national serological survey [21,22]. We

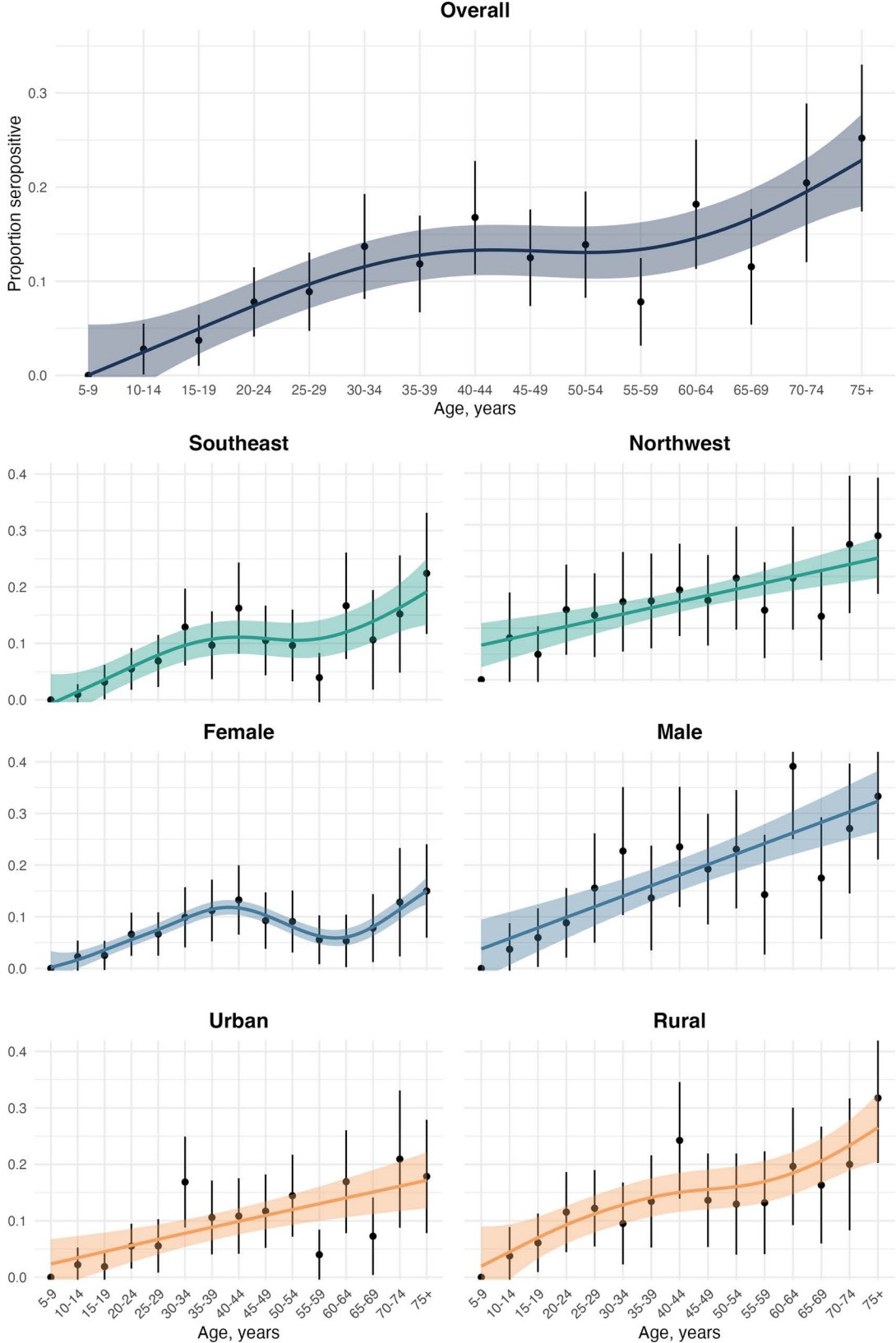

**Fig 3. Age-stratified seroprevalence by region, gender and urban vs. rural setting.** Individual plots show seroprevalence point estimates (black dots) and 95% confidence intervals (black vertical lines) by five-year age group. A smoothed line is fitted

to the data using generalized additive models with a cubic spline and weighted for number of observations. Colored ribbons represent the 95% CI. Number of observations per plot listed in Table 1. Southeast study region (San Pedro de Macoris province). Northwest study region (Espaillat province). Additional plots showing age stratified seroprevalence by both gender and study region are presented in S2 Fig.

examined both the overall epidemiology of human *Leptospira* infection and the epidemiology of the three most common serogroups: Icterohaemorrhagiae, Australis, and Canicola. Icterohaemorrhagiae, primarily transmitted by exposure to rat urine, was the most commonly identified serogroup. Given Icterohaemorrhagiae is considered to cause the most severe clinical disease, including Weil's Disease which is characterized by hepatic and renal failure and high case-fatality rates, it is of particular public health importance [23]. The primary reservoirs for Icterohaemorrhagiae are the *Rattus norvegicus* (or brown rat) and *Rattus rattus* (Black rat), while other rodents including mice and voles may be involved in transmission. Australis and Canicola serogroups are usually associated with milder clinical disease, although severe disease and deaths have been reported. The primary reservoirs for the Australis serogroup include various wild and domestic animals, such as farm animals, rats, and mice. Dogs are the major recognized reservoir for the Canicola serogroup, although livestock and wild mammal are other potential reservoirs [24].

Similar to studies on the epidemiology of leptospirosis from other settings, we confirmed that being male, being exposed to rats in the home, and older age were associated with higher seroprevalence [19]. We found that the prevalence of *Leptospira* antibodies increased through middle age and then plateaued, a typical finding that reflects the increased occupational risk of infection among young and middle aged adults. However, we also observed an increase in the oldest age groups, particularly among males, a somewhat unusual trend given reported infection rates generally decline among older age groups [1], with seroprevalence typically remaining flat or declining during late adulthood [19]. Given antibodies to *Leptospira* decay over time and are thought to be detectable for only about eight years post-infection [3,25], the older population would be anticipated to register declining seroprevalence in the absence of ongoing exposures. The reason that this was not the case suggests unexpectedly high ongoing transmission among older adults, although why age-specific transmission would be different in this versus other settings is unclear. We further explored age-specific seroprevalence stratifying by gender, study region and urban vs rural setting and found interesting differences, particularly between males and females. Males demonstrated a near linear increase in seroprevalence with age while females demonstrated a distinct biphasic pattern with peaks around early middle-age and again after 70 years of age. The reason for these differences by gender is unclear and the few studies that have reported *Leptospira* seroprevalence stratified by both age and gender (or sex) contrast with our findings, although those studies were either limited by size [26] or from markedly different settings [19]. We also found differences by study region, with seroprevalence in the Northwest Region (Espaillat) about 1.8-fold higher than the Southeast Region (San Pedro de Macoris) even after adjusting for potential confounders, highlighting distinct spatial heterogeneity in leptospirosis transmission. However, there are nuances to these findings, and there were differences in serogroup-specific risks that were not identified when assessing risks aggregated across all serogroups. Given differences in mammal reservoirs across serogroups, and therefore differences in exposure risks and transmission pathways, this may be anticipated. For example, adjusted serogroup Australis seroprevalence was about five-fold higher among farmers than the reference group, but not higher for Icterohaemorrhagiae or Canicola. This likely reflects the fact that livestock is a common reservoir for the Australis serogroup but less so for Icterohaemorrhagiae and Canicola. Similarly, the adjusted odds

of being seropositive for Icterohaemorrhagiae was almost three-fold higher for males than females, while no gender differences were identified for serogroups Australis and Canicola. These data highlight how serogroup-specific risks may be leveraged to prioritize targeted public health measures such as rodent control or livestock and dog vaccination [27].

While our study has multiple strengths including a relatively large population selected using a multilevel sampling approach, testing of samples using the reference-standard serological assay, and sampling across two discrete regions to understand geographic variations, there are some limitations. First, the geographic scope is limited to two out of 34 provinces, potentially restricting the generalizability of our findings across other regions of the country. Second, cross-reactivity may affect the accuracy of our serogroup-specific analyses. However, when we excluded seropositive individuals with highest titer against a serogroup other than the one being assessed, findings were essentially unchanged. Irrespective, given well established limitations of MAT for definitely identifying the infecting serogroup, and given we were reporting putative serogroups based on the highest reacting titer, our serogroup-specific findings should be considered as suggestive rather than definitive. Third, while some studies use an MAT titer cutoff of 1:50, we used a titer cutoff of 1:100 and therefore our estimated seroprevalence was likely lower than if a lower cutoff was used. Fourth, self-reported exposures and socio-demographics factors enumerated through our survey may be subject to recall or social desirability biases. Fifth, serogroup specific analyses were limited by the number of seropositive individuals, particularly for serogroups Australis and Canicola. Lastly, this was a cross-sectional study and therefore we were not able to capture temporal transmission dynamics.

In conclusion, this seroepidemiological study provides insights into the epidemiology and under-reported burden of human *Leptospira* infection in the Dominican Republic. We identified a range of serogroups, geographic variation in infection and risk factors, and a substantial discrepancy between reported cases and likely infections. While seroprevalence is high, risks varied markedly by age, gender, and study region. Analyzing serogroups individually suggested distinct transmission patterns, emphasizing the importance of understanding transmission pathways to ensure targeted and effective public health interventions. Our findings highlight the complex epidemiology of leptospirosis in the Dominican Republic, underlining the need for more granular investigation into environmental factors, animal reservoirs, and serovar-specific dynamics to effectively control this pathogen.

## Supporting information

**S1 Table. Seroprevalence of *Leptospira* by primary reacting serovar, Espaillat and San Pedro de Macoris Provinces, Dominican Republic, July-Oct 2021.** Primary reacting serovar is defined as the serovar that registered the highest titer on microscopic agglutination test (MAT) for each seropositive study participant. Individuals that registered the same highest titer to two or more serovars are categorized as 'mixed.' Numbers appear to be inconsistent between Table 2 and this table given some individuals included in the 'Mixed' category in this table had the highest titers to two serovars from the same serogroup. For example, 13 individuals categorized as 'Mixed' in this table registered the highest titers to serovars Mankarso and Icterohaemorrhagiae; given these are both serogroup Icterohaemorrhagiae they are categorized by serogroup rather than 'Mixed' in Table 2.
(DOCX)

**S2 Table. Priors, model estimates and DIC for serocatalytic models.** FOI, force of infection; DIC, deviance information criterion.
(DOCX)

**S3 Table. National reported leptospirosis cases, Dominican Republic 2013–2022.** Data on national reported leptospirosis cases was from the Pan American Health Organization (PAHO) Core Indicator Dashboard from data generated by the General Epidemiology Directorate of the Ministry of Public Health and Social Assistance.
(DOCX)

**S4 Table. Odds ratios for testing seropositive for any *Leptospira* serogroup, Espaillat and San Pedro de Macoris Provinces, Dominican Republic, July-Oct 2021.** Number in dataframe = 2091, Number in model = 2089, Missing = 2, AIC = 1367.8, C-statistic = 0.721, H&L = Chi-sq(8) 1.93 (p = 0.983). N = 237 seropositive cases. San Pedro de Macoris province (Southeast study region). Espaillat province (Northwest study region). Seropositive defined as ≥ 1:100 titers using the microscopic agglutination test.
(DOCX)

**S5 Table. Odds ratios for testing seropositive for *Leptospira interrogans* serogroup Icterohaemorrhagiae, Espaillat and San Pedro de Macoris Provinces, Dominican Republic, July-Oct 2021.** Number in dataframe = 2091, Number in model = 2089, Missing = 2, AIC = 701.2, C-statistic = 0.751, H&L = Chi-sq(8) 5.53 (p = 0.699). N = 91 seropositive cases. San Pedro de Macoris province (Southeast study region). Espaillat province (Northwest study region). Seropositive defined as ≥ 1:100 titers using the microscopic agglutination test.
(DOCX)

**S6 Table. Odds ratios for testing seropositive for *Leptospira interrogans* serogroup Australis, Espaillat and San Pedro de Macoris Provinces, Dominican Republic, July-Oct 2021.** Number in dataframe = 2091, Number in model = 2089, Missing = 2, AIC = 364.5, C-statistic = 0.746, H&L = Chi-sq(8) 3.04 (p = 0.932). N = 38 seropositive cases. San Pedro de Macoris province (Southeast study region). Espaillat province (Northwest study region). Seropositive defined as ≥ 1:100 titers using the microscopic agglutination test.
(DOCX)

**S7 Table. Odds ratios for testing seropositive for *Leptospira interrogans* serogroup Canicola, Espaillat and San Pedro de Macoris Provinces, Dominican Republic, July-Oct 2021.** Number in dataframe = 2091, Number in model = 2089, Missing = 2, AIC = 264, C-statistic = 0.781, H&L = Chi-sq(8) 3.01 (p = 0.934). N = 25 seropositive cases. Reference category for age group 20–34 vs 5–19 for prior models given absence of seropositive cases in the youngest age category. San Pedro de Macoris province (Southeast study region). Espaillat province (Northwest study region). Seropositive defined as ≥ 1:100 titers using the microscopic agglutination test. NC, not calculated.
(DOCX)

**S8 Table. Multicollinearity of model covariates.** Generalized variance inflation factor (GVIF) and degrees of freedom (DF) for each covariate included in the regression models. The adjusted GVIF accounts for the degrees of freedom of each covariate to simplify interpretation across covariates with different degrees of freedom. An adjusted GVIF value close to 1 indicates multicollinearity is not substantially inflating the variance of the estimated regression coefficients for that covariate.
(DOCX)

**S1 Fig. Serocatalytic models fitted to age stratified *Leptospira* seroprevalence by microscopic agglutination test, Dominican Republic 2021.** (A) Catalytic model. (B) Reverse catalytic model. (C) Reverse catalytic model with no transmission in the six years prior to the serosurvey.
(PNG)

**S2 Fig. Age-stratified *Leptospira* seroprevalence by microscopic agglutination test, by region and gender, Dominican Republic 2021.** Individual plots show seroprevalence point estimates (black dots) and 95% confidence intervals (black vertical lines) by five-year age group. A smoothed line is fitted to the data using generalized additive models with a cubic spline and weighted for number of observations. Colored ribbons represent the 95% CI. (JPG)

**S3 Fig. Association between age and seroprevalence, Dominican Republic 2021.** Generalized Additive Models indicate the proportion of individuals with *Leptospira* antibodies (titers ≥1:100), stratified by gender. The models included smooth terms for age to capture non-linear age effects and adjusted for study region, residing in a barrio (settlement), work environment, and rural vs. urban residence. Ribbons represent the 95% CI. (JPG)

## Acknowledgments

We would like to thank the many study participants that volunteered to participate in this study. We would also like to thank the study staff that collected the field data, the Dominican Republic Ministry of Health and Social Assistance, and the Pedro Henriquez Ureña National University, for their commitment and support for the study. The findings and conclusions in this report are those of the authors and do not necessarily represent the official position of the US CDC.

## Author contributions

**Conceptualization:** Eric J. Nilles, Adam Kucharski, Colleen L. Lau.

**Formal analysis:** Eric J. Nilles, Eleanor M. Rees.

**Funding acquisition:** Eric J. Nilles, Adam Kucharski, Colleen L. Lau.

**Investigation:** Cecilia Then Paulino, Renee Galloway, Michael de St. Aubin, Helen J. Mayfield, Angela Cadavid Restrepo, Devan Dumas, Salome Garnier, Marie Caroline Etienne, William Duke, Farah Peña, Naomi Iihoshi, Gabriela Abdalla, Beatriz Lopez, Lucia de la Cruz, Bernarda Henríquez, Kara Durski.

**Methodology:** Cecilia Then Paulino.

**Project administration:** Cecilia Then Paulino, Devan Dumas, Margaret Baldwin, Ronald Skewes-Ramm, Emily Zielinski Gutiérrez.

**Supervision:** Eric J. Nilles, Cecilia Then Paulino, Devan Dumas, Adam Kucharski, Colleen L. Lau.

**Visualization:** Eric J. Nilles.

**Writing – original draft:** Eric J. Nilles.

**Writing – review & editing:** Cecilia Then Paulino, Renee Galloway, Michael de St. Aubin, Helen J. Mayfield, Angela Cadavid Restrepo, Devan Dumas, Salome Garnier, Marie Caroline Etienne, William Duke, Farah Peña, Naomi Iihoshi, Gabriela Abdalla, Beatriz Lopez, Lucia de la Cruz, Bernarda Henríquez, Kara Durski, Margaret Baldwin, Gideon Loevinsohn, Eleanor M. Rees, Beatris Martin, Benn Sartorius, Ronald Skewes-Ramm, Emily Zielinski Gutiérrez, Adam Kucharski, Colleen L. Lau.

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
