## [Decision Letter · Decision Letter 0]

20 Nov 2024

Response to Reviewers
Revised Manuscript with Track Changes
Manuscript

Shaden Kamhawi

co-Editor-in-Chief

Paul Brindley

co-Editor-in-Chief

**Journal Requirements:**

At this stage, the following Authors/Authors require contributions: Eric J. Nilles, Cecilia Then Paulino, Renee Galloway, Michael de St, Aubin, Helen J. Mayfield, Angela Cadavid Restrepo, Devan Dumas, Salome Garnier, Marie Caroline Etienne, William Duke, Farah Peña, Naomi Iihoshi, Gabriela Abdalla, Beatriz Lopez, Lucia de la Cruz, Bernarda Henríquez, Kara Durski, Gideon Loevinsohn, Eleanor M. Rees, Beatris Martin, Benn Sartorius, Ronald Skewes-Ramm, Emily Zielinski Gutiérrez, Adam Kucharski, and Colleen L. Lau. Please ensure that the full contributions of each author are acknowledged in the "Add/Edit/Remove Authors" section of our submission form.

2) Please provide an Author Summary. This should appear in your manuscript between the Abstract (if applicable) and the Introduction, and should be 150u2013200 words long. The aim should be to make your findings accessible to a wide audience that includes both scientists and non-scientists. Sample summaries can be found on our website under Submission Guidelines:

3) Your manuscript is missing the following sections: Introduction.  Please ensure all required sections are present and in the correct order. Make sure section heading levels are clearly indicated in the manuscript text, and limit sub-sections to 3 heading levels. An outline of the required sections can be consulted in our submission guidelines here:

5) We have noticed that you have uploaded Supporting Information files, but you have not included a list of legends. Please add a full list of legends for your Supporting Information files after the references list.

6) Please amend your detailed Financial Disclosure statement. This is published with the article. It must therefore be completed in full sentences and contain the exact wording you wish to be published. Please ensure that the funders and grant numbers match between the Financial Disclosure field and the Funding Information tab in your submission form. Note that the funders must be provided in the same order in both places as well.

**Reviewers' comments:**

**Key Review Criteria Required for Acceptance?**

**Methods**

-Are the objectives of the study clearly articulated with a clear testable hypothesis stated?

-Is the study design appropriate to address the stated objectives?

-Is the population clearly described and appropriate for the hypothesis being tested?

-Is the sample size sufficient to ensure adequate power to address the hypothesis being tested?

-Were correct statistical analysis used to support conclusions?

-Are there concerns about ethical or regulatory requirements being met?

Reviewer #1: The manuscript describes a seroepidemiological study of leptospirosis in the Dominican Republic. Due to the lack of laboratory diagnosis and awareness, leptospirosis is underdiagnosed, and surveillance data underestimates its true impact. This study is an important contribution, demonstrating the role of seroepidemiological investigations in populations with limited knowledge of leptospirosis. It is a cost-effective approach to obtain valuable epidemiological information, generate hypotheses, and inform improvements in diagnosis and surveillance systems.

**Results**

-Does the analysis presented match the analysis plan?

-Are the results clearly and completely presented?

-Are the figures (Tables, Images) of sufficient quality for clarity?

Reviewer #1: The manuscript reinforces the fact that leptospirosis is underestimated and provides a good case study and methodological details that can be useful for others. However, some important limitations should be considered and addressed:

- MAT is the standard methodology for the determination of Leptospira-specific antibodies and positivity means past exposure. Authors hint at MAT's limitation in identifying infecting serogroups but still analyzed the data, interpreted results, and made conclusions as it reflects circulating serogroups. For the serogroup sub-analysis, I appreciated they conducted a sensitivity analysis but the report should elaborate more on the limitation and make clear that they are simply reporting serogroups with the highest reacting titers. They cannot discuss results or make conclusions about circulating serogroups. Any comparison with other studies should reflect this as well.

- I appreciate the effort to estimate FOI and inferences about the expected number of infections, including examining different models and assumptions. It is recognized that many infections are missed; this exercise demonstrates that. Authors inferred 33,000 infections per year, much higher than the reported counts; however, authors are comparing expected infections with the number of people who were ill enough to seek care, then tested, and reported. I don’t think those are comparable. The spectrum of leptospirosis illness is broad and a fraction of people may have subclinical disease, self-limiting mild illness, etc. and therefore would never make it into the reportable system. Others are ill but not cannot access care. Others seek care but they are not tested. The paper should be clear about the different comparisons they want to make, their purpose, and impact.

- Authors conducted various analyses of sub-groups, and found highest rates among the older population. There was no attempt to explain this finding by providing some ideas for behaviors or exposure that could explain the findings. For example, is it that people at those ages may switch to spending more time at home and there is an increase in domestic exposures?

- MAT titers are only displayed in a figure and not mentioned in the results and discussion. MAT titers can provide additional information about the transmission dynamics. It seems like most are low titers, 1:100. Results can highlight the profiles of those with high titers.

**Conclusions**

-Are the conclusions supported by the data presented?

-Are the limitations of analysis clearly described?

-Do the authors discuss how these data can be helpful to advance our understanding of the topic under study?

-Is public health relevance addressed?

Reviewer #1: (No Response)

**Editorial and Data Presentation Modifications?**

Reviewer #1: (No Response)

**Summary and General Comments**

Reviewer #1: (No Response)

PLOS authors have the option to publish the peer review history of their article (what does this mean? ). If published, this will include your full peer review and any attached files.

**Do you want your identity to be public for this peer review?** For information about this choice, including consent withdrawal, please see our Privacy Policy .

Reviewer #1: No

**Figure resubmission:****Reproducibility:** To enhance the reproducibility of your results, we recommend that authors of applicable studies deposit laboratory protocols in protocols.io, where a protocol can be assigned its own identifier (DOI) such that it can be cited independently in the future. Additionally, PLOS ONE offers an option to publish peer-reviewed clinical study protocols. Read more information on sharing protocols at https://plos.org/protocols?utm_medium=editorial-email&utm_source=authorletters&utm_campaign=protocols

---

## [Editor Report · Decision Letter 1]

3 Dec 2024

Dear Dr. Nilles,

We are pleased to inform you that your manuscript 'Seroepidemiology of human leptospirosis in the Dominican Republic: a multistage cluster survey, 2021' has been provisionally accepted for publication in PLOS Neglected Tropical Diseases.

Best regards,

Ana LTO Nascimento

Section Editor

Ana LTO Nascimento

Section Editor

Shaden Kamhawi

co-Editor-in-Chief

Paul Brindley

co-Editor-in-Chief

The authors have answered all issues raised by reviewer.

---

## [Editor Report · Acceptance letter]

Dear Dr Nilles,

We are delighted to inform you that your manuscript, "Seroepidemiology of human leptospirosis in the Dominican Republic: a multistage cluster survey, 2021," has been formally accepted for publication in PLOS Neglected Tropical Diseases.

Best regards,

Shaden Kamhawi

co-Editor-in-Chief

Paul Brindley

co-Editor-in-Chief
